# Survival Analysis of Symptomatic COVID-19 in Phuentsholing Municipality, Bhutan

**DOI:** 10.3390/ijerph182010929

**Published:** 2021-10-18

**Authors:** Kinley Gyeltshen, Tsheten Tsheten, Sither Dorji, Thinley Pelzang, Kinley Wangdi

**Affiliations:** 1Phuentsholing Hospital, Chukha 21102, Bhutan; sitherdorji059@gmail.com (S.D.); tpelzang2015@gmail.com (T.P.); 2Department of Global Health, Research School of Population Health, Australian National University, Canberra, ACT 2602, Australia; tsheten.tsheten@anu.edu.au (T.T.); kinley.wangdi@anu.edu.au (K.W.)

**Keywords:** COVID-19, patients, Bhutan, survival, analysis, symptomatic

## Abstract

COVID-19 is a disease that is caused by a highly transmissible and pathogenic novel coronavirus: severe acute respiratory syndrome coronavirus 2 (SARS-CoV-2). All of the COVID-19 positive cases in Bhutanese travellers returning via the Phuentsholing point of entry, the local population, and Indian nationals were isolated in the Phuentsholing COVID-19 isolation ward, Bhutan. This study aimed to identify the risk factors for developing symptoms among COVID-19 positive patients in this ward. A retrospective cohort study was conducted using the data regarding COVID-19 positive cases in the Phuentsholing COVID-19 isolation ward from 28 May 2020 to 31 May 2021. The Cox proportional hazards regression model was used to identify the risk factors of developing COVID-19 symptoms. There were 521 patients in the study; 368 (70.6%) were males and 153 (29.4%) were females. The mean age was 32 years (with a range of 1–78 years), and 290 (56.0%) reported at least one symptom. The median length of isolation was eight days (with a range of 3–48 days). The common symptoms were: cough (162, 31.0%), fever (135, 26.0%), and headache (101, 19.0%). In the multivariable Cox regression, vaccinated patients were 77.0% (*p* = 0.047) less likely to develop symptoms compared to those who were not vaccinated. The front line workers and the mini-dry port (MDP) workers were 15 (*p* = 0.031) and 41 (*p* < 0.001) times more likely to be symptomatic compared to returning travellers. The young and economically active population group was most commonly affected by COVID-19. The presence of risk factors, such as being front line workers, MDP workers, or not being vaccinated against COVID-19, meant that patients had a higher probability of developing symptoms of COVID-19.

## 1. Background

Coronavirus disease 19 (COVID-19) emerged in Wuhan, Hubei Province, China in late December 2019, associated with the highly transmissible and pathogenic novel coronavirus: severe acute respiratory syndrome coronavirus 2 (SARS-CoV-2) [1]. SARS-CoV-2 is continuing to spread, with 228 million cases and more than four million deaths recorded worldwide as of 19 September 2021 [2]. In response, many countries were forced to implement extreme measures, such as physical distancing, quarantine, and lockdowns. These responses had serious economic, social, and psychological effects on the population [3,4]. The burden on both curative and preventive healthcare sectors significantly increased, particularly in developing countries with limited resources [5].

The most common symptoms in the early stage of the disease include a fever (98%), dry cough (76%), myalgia (44%), and fatigue (44%) [6]. Among these symptoms, fatigue and sputum were identified as signs of severe COVID-19 infection [7]. Whilst a dry cough is commonly observed in the early stage of the disease, the production of sputum becomes more prominent as the disease progresses [8]. Older patients (>60 years) and those with chronic diseases were more likely to have severe health outcomes, poor prognosis, and were at greater risk of dying [9]. What is even more worrying is that there is currently no approved treatment for COVID-19, and there is little, if any, clinical trial data that support any prophylactic treatments [10].

Although the COVID-19 pandemic has been ongoing for nearly three years, there is a paucity of literature that distinguishes the risk factors of symptomatic and asymptomatic COVID-19 patients. Moreover, the effects of COVID-19 vaccinations on the development of symptoms are still unclear. Understanding the factors that lead to symptom manifestation in COVID-19 patients is useful for providing insight into disease severity, as well as for resource allocation and workforce mobilization.

Bhutan started its first nationwide COVID-19 vaccination campaign in March 2021. The campaign resulted in a vaccination coverage of >94% of its eligible population [11,12]. The Oxford–AstraZeneca (Covishield) vaccine, which is a replication-defective chimpanzee adenoviral vector expressing the SARS-CoV-2 spike glycoprotein (also known as ChAdOx1), was used in the first national vaccine rollout campaign. In one of the clinical trials, the efficacy of this vaccine was reported to be 90% in participants who received a low dose (2.2 × 10^10^ viral particles) followed by a standard dose (5 × 10^10^ viral particles) [13].

Bhutan also recorded a surge in cases, particularly in the Phuentsholing municipality and other southern districts, where the Delta variant of SARS-CoV-2 was detected in >80% of these cases (personal communication). Real-world data showed that the ChAdOx1 and BNT162b2 vaccines have similar efficacies after one shot, with protection rates of 33.5% and 51.1% against the Alpha and Delta variants, respectively. However, after two shots, the efficacy of ChAdOx1 against the Alpha and Delta variants was 66.1% and 59.8%, respectively [14,15]. Therefore, there is a need to understand the epidemiological characteristics of the disease in such hotspots to inform policymakers to make the right decisions regarding the ongoing COVID-19 pandemic.

Since the first case of COVID-19 on 5 March 2020, 2596 cases and three deaths have been recorded (as of 6 September 2021) in Bhutan [16]. There have been several instances of local transmission associated with the incoming travellers and uncontrolled human movement in the southern districts bordering India [17,18]. Additionally, the illegal movement of people across the international border heightened the risk of SARS-CoV-2 infection [19].

Understanding the severity of the disease in relation to the clinical manifestation of COVID-19 is paramount to enrich the available data and guide the clinical management of this new disease. This study aimed to identify the host factors that predispose them to manifest COVID-19 symptoms with SARS-CoV-2 infection.

## 2. Methods

### Study Setting

The study was conducted in the Phuentsholing municipality of Chukha District. Of the 11 sub-districts in Chukha District, the Phuentsholing municipality in the Phuentsholing sub-district is the most densely populated area, where >40% of the district population lives. The 2017 population housing census of Bhutan (PHCB) recorded 27,658 individuals in Phuentsholing, excluding non-Bhutanese and tourists [20].

The Phuentsholing municipality is the commercial hub of Bhutan. Most of the goods and commodities (including groceries, clothes, vegetables, construction materials, and building materials) are ferried through Phuentsholing from India to other places in the country [21]. It shares an international boundary with the Indian Town of Jaigaon in West Bengal (Figure 1). People from both sides can freely migrate from either side of the border via the point of entry (PoE) designated by the two countries. Most settlements reside in close proximity and are only separated by an international border (wall).

## 3. Study Design and Study Population

This research involved a retrospective cohort study that included laboratory-confirmed COVID-19 positive individuals from the isolation wards of Phuentsholing municipality between 27 May 2020 and 31 May 2021. All patients tested positive in real-time reverse transcriptase-polymerase chain reaction (RT-PCR) tests. The patients were observed for any symptoms during the entire course of the isolation period. Follow-up ended on the day of the onset of symptoms or the day of discharge, whichever occurred first. Hence, the study comprised symptomatic and asymptomatic cases (i.e., symptomatic vs. asymptomatic), with the time until the onset of symptoms as the primary outcome. The time of the onset of symptoms was calculated as the date of the onset of symptoms minus the date of isolation in the ward. Patients were discharged when they tested negative for RT-PCR on the 5th day of isolation. The test was repeated every 48–72 h until it was negative, as per the local guideline. These patients formed the censoring group in our analysis, who were symptom-free and tested negative for COVID-19 at the end of the isolation period. The final sample constituted patients that were both unvaccinated and vaccinated against SARS-CoV-2 infection. The vaccinated group in this study had only received one dose of the SARS-CoV-2 vaccine at the time of the study.

The inclusion criteria were patients with: (1) RT-PCR positive COVID-19; (2) all individuals admitted to Phuentsholing isolation ward without restriction of age and sex; (3) asymptomatic before or on the day of admission to the ward. Exclusion criteria were: (1) patients that developed symptoms before or during the day of isolation; (2) missing date of onset of symptoms.

## 4. Data Source

Data for the study was obtained from the registry of the COVID-19 isolation ward in the Phuentsholing municipality. The variables extracted were: age, sex, occupation, vaccination status, comorbidities, clinical symptoms, date of admission, date of onset of symptoms, and date of discharge. The data were coded and entered into Epi-data Entry version 3.1 (EpiData Association, Odense, Denmark, http://www.epidata.dk/ Accessed on 10 June 2021). Data were entered by two independent investigators and errors were corrected.

The sources of COVID-19 patients in Phuentsholing were: (1) the flu clinic; (2) quarantine facilities of the returning travellers and foreign nationals entering Bhutan via Phuentsholing PoE; (3) workers of mini-dry port (MDP) who deal with imports and exports; (4) community-based mass testing and contact tracing of the active cases. Each of these data sources are described here under.

Immediately after COVID-19 was declared as a public health emergency of international concern by the World Health Organization (WHO), a flu clinic was set up in the municipality for the early identification of COVID-19 cases by the Ministry of Health. Any patients visiting this flu clinic that were suspected to have COVID-19 (based on clinical and epidemiological investigation) were required to collect their samples to test for SARS-CoV-2. For the first time, a molecular laboratory was set up in the Phuentsholing General Hospital to perform RT-PCR, in line with the national strategy for the early detection and control of COVID-19 in the southern region. This meant that COVID-19 samples from the neighbouring hospitals in the region had to be referred to this hospital instead of being shipped to the national reference laboratory in the capital. According to the national pandemic preparedness response guidelines, any travellers entering Bhutan via the Phuentsholing PoE must undergo a mandatory 21-day quarantine period at designated facilities at the expense of the government. During this period, all travellers were monitored for any signs and symptoms of COVID-19 and tested for SARS-CoV-2. Phuentsholing MDP is one of the busiest centres dealing with trade—Indian lorries bring in imported items, some of which are essential goods. COVID-19 outbreaks occurred in this place in the past due to the casual contact of Indian drivers and Bhutanese workers during transportation. Following the sporadic detection of COVID-19 cases in Phuentsholing, community-based mass SARS-CoV-2 testing was conducted on several occasions in 2020. In 2021, three such mass testing programs were carried out (the latest being carried out between June 10 and June 16, 2021). During this mass testing, at least one member in the household was tested for SARS-CoV-2. Following the confirmation of COVID-19, all close contacts related to the index case(s) were identified by the contact tracing team and were required to complete a 21-day quarantine. Finally, all confirmed COVID-19 cases were admitted to the isolation centre in the municipality for further observation and management. An electronic map of municipalities in shapefile format was obtained from the DIVA-GIS database (https://www.diva-gis.org/ Accessed on 10 June 2021).

## 5. Statistical Analysis

Descriptive analysis was conducted using frequencies and proportions for categorical variables, and means and standard deviations (SD) for continuous variables. Symptomatic cases were further categorized into three groups:Mild disease: those with at least one symptom of COVID-19 (fever, fast breathing, cough, and chest x-ray (CXR) change, such as consolidation) but no signs of pneumonia;Severe disease: in addition to all or some COVID-19 symptoms, this group of patients had pneumonia (evidenced by CXR changes), needing antibiotics and oxygen therapy;Life-threatening: patients who had severe pneumonia and were haemodynamically unstable, needing intensive care unit admission.

The available haematological and serological data were obtained, and the mean and SD were calculated for these variables. The mean duration at which the antibodies were detected was calculated for some patients.

Survival analysis was carried out to estimate the cumulative probability of the onset time of symptoms for each risk factor. Kaplan–Meier curves were used to estimate the probabilities for the onset of symptoms at different time points, and a log rank test was employed to assess the differences between covariates [22]. Here, survival time corresponds to the total number of days elapsed between the date of isolation and the date of onset of symptoms (hazard). The Cox proportional regression model was used to estimate hazard ratios (HR) and the corresponding 95% confidence intervals for symptomatic cases for all possible risk factors (including age, sex, vaccination status, and occupational group).

A *p*-value of <0.05 was considered statistically significant. Analysis was carried out using STATA Version 13 (StataCorp, Stata Statistical Software, licensed Khesar Gyalpo University, Royal University of Bhutan). The map was created using ArcMap 10.5.1 software (ESRI, Redlands, CA, USA).

## 6. Results

### Descriptive Analysis

A total of 521 patients with RT-PCR positive COVID-19 were admitted to the COVID-19 isolation ward of Phuentsholing from 28 May 2020 to 31 May 2021. The majority were males (male vs. female: 70.6% vs. 29.4%) and the mean age was 32 years (range: 1–78 years). More than half of them (52.4%) were in the age group of 19–35 years. Uncomplicated diabetes mellitus was the most common comorbidity, being present in 12 cases (2.3%). Of 290 (56.0%) symptomatic cases, 288 (98.0%) had mild symptoms (indicative of mild disease). One hundred and ninety-one (36.7%) had received the first shot of the COVID-19 vaccine.

Three-fourths (*n* = 393) of the total patients were Bhutanese and the remaining (n = 129) were Indian. Among the Indian patients, 111 (86.7%) were expatriate workers and the rest were Indian Military Training Team(IMTRAT) and Border Road Organization (DANTAK) personnel (*n* = 17). Two-hundred-and-thirty-four (45.0%) COVID-19 positive cases were related to the community transmission through their primary contacts. There were 81 (16.0%) Bhutanese returnees that had been repatriated from other countries in the isolation ward. The rest were non-health frontline workers (Figure 2). The frontline workers constituted army, police, foresters, customs, and *desuups* (civilian volunteers who are trained to provide relief support during disasters).

Private/corporate/business employees were the most common occupational group, with 200 cases (38.4%), followed by 78 (15.0%) students, including those who returned from India. Farmers and armed forces/uniformed personnel consisted of 67 (12.9%) and 65 (12.5%) cases, respectively. Homemakers, monks and nuns, and civil servants comprised the remaining cases (Table 1). The overall median length of stay in isolation was eight days (SD: 6.09, range 3–48 days).

Among the patients with symptoms, 290 (56.0%) experienced at least one symptom. The most commonly reported symptoms were a cough (*n* = 162, 31.0%), fever (*n* = 135, 26.0%), and headache (*n* = 101, 19.0%). Other symptoms, including a sore throat (*n* = 82, 16.0%), myalgia (*n* = 63, 12%), runny nose (*n* = 59, 9.0%), loss of smell sensation (*n* = 42, 8.0%), and loss of taste sensation (*n* = 34, 7.0%), were also reported. Diarrhoea, abdominal pain, nausea, and shortness of breath were infrequently reported and constituted <5% (Figure 3).

The mean white cell count on admission was 6.52 ± 2.149 × 10^3^ per microliter; the mean absolute lymphocyte count was 1.88 ± 7540 × 10^3^ per microliter; and the mean platelet count was 239.72 ± 86.546 × 10^3^ per microliter. The mean duration at which the IgM antibodies against COVID-19 were detected on the rapid test was 17.4 days after admission and 18.5 for IgG antibodies (Table 2).

## 7. Survival Analysis

Of the 521 isolated patients, 214 (41.1%) were symptomatic on admission and 47 (9.0%) developed symptoms on the day of admission and were excluded from survival analysis. Therefore, 258 records (excluding two with the wrong date of symptom onset) were included for the final analysis. The Kaplan–Meier curve and log rank test showed that there was no significant difference (*p* = 0.504) in the probability of the occurrence of COVID-19 (with symptoms) between the male and female sex. However, a significant difference was observed in the Kaplan–Meier curve regarding the origin of cases, the presence of risk factors, and the category of absolute lymphocyte count <1000/mL (*p* < 0.001) (Figure 4).

These results were further supplemented by the significant differences in the logrank tests for origin of cases (*p* < 0.001) and presence of risk factors (*p* < 0.001) for absolute lymphocyte count (*p* = 0.022) (Figure 5).

In the multivariable Cox proportional hazard model, the COVID-19 vaccinated patients were 77.0% less likely to develop COVID-19 symptoms compared to those that were not vaccinated (adjusted hazard ratio (aHR): 0.23; 95% CI: 0.05, 0.98). Conversely, compared to travellers returning from abroad, the frontline workers and the MDP workers were 14.90 times (aHR: 14.9; 95% CI: 1.27, 174.09) and 41.29 times (aHR: 41.29; 95% CI: 4.24, 402.36) more likely to develop COVID-19 symptoms, respectively. However, no significant differences were observed in the risk of developing symptomatic COVID-19 infection among the different age group, occupation, and sex variables (Table 3).

## 8. Discussion

This study revealed that the majority of COVID-19 patients were in the age group of 19–35 years old. One-third of the cases worked in private companies, businesses, or corporations. The most commonly reported symptoms were a cough, fever, headache, sore throat, and runny nose. In the multivariable Cox proportional hazards regression model, vaccinated individuals were less likely to develop symptoms compared to those who were not vaccinated. MDP workers and frontline workers were more likely to develop symptoms compared to Bhutanese returnees from abroad.

The majority of the affected patients belonged to the young and economically active population in the age group of 19–35 years (median age: 32 years). The possible reason for this could be that younger people are more likely to travel for work compared to older people. The median age of the COVID-19 patients in this study was much younger compared to other studies (>40 years) [23]. This difference might be related to different socio-demographic settings, which underscores the need to understand the local characteristics of the disease to inform the location-specific decision-making process.

Among the occupational categories, more than one third of COVID-19 patients were private and business workers. This is the case because the Phuentsholing municipality is the place in which many Bhutanese entrepreneurs and private workers operate their businesses. The Phuentsholing municipality provides the main platform for trade between Bhutan and India. Therefore, these groups of people are at higher risk for SARS-CoV-2 infection as they are more frequently involved in interaction and travel. Previous studies also suggested that COVID-19 cases could be related to occupational exposure in a significant proportion of cases [24,25].

In this study, the median length of isolation stay was eight days. This finding is similar to what was found in the United States and several European countries, where the duration of hospital stay was 7–8 days on average [26,27,28]. Our findings on the clinical manifestations of COVID-19 (including a cough, fever, and headache) were different from previous studies [29,30]. In another study, >90% of symptomatic patients had a fever, 50–76% patients reported a cough, and 25.3–44% of the patients suffered fatigue [31]. In addition, patients that were admitted to Phuentsholing isolation ward also presented symptoms such as a sore throat, myalgia, runny nose, anosmia, and a loss of taste. Diarrhoea, abdominal pain, nausea, and shortness of breath were among the other symptoms that were noticed. This is supported by a systematic review that reported 27 signs and symptoms, which can be categorized into four different groups: systemic, respiratory, gastrointestinal, and cardiovascular [32]. Similar symptoms were reported in China and the cases were primarily mild to moderate diseases [33].

The risk of being symptomatic was significantly reduced in vaccinated people. Although Bhutan has completed two rounds of the vaccination campaign, at the time of this study, only one round of vaccination had been completed. Similar to our findings, other studies have also reported a decline in disease incidence among the vaccinated workers [34]. Vaccines are effective for both the Alpha and Delta variants of SARS-CoV-2 [35]; there was a significant reduction in symptomatic COVID-19 in older adults, and protection against severe disease amongst those who were vaccinated [36]. However, in Israel, the effectiveness of the first dose of the COVID-19 vaccination against symptomatic COVID-19 was only 54% [37].

The frontline workers, comprising *desuups* (volunteers), police, and armies, were 15 times more at risk of developing COVID-19 symptoms compared to Bhutanese nationals returning from other countries. Frontline workers are involved in numerous tasks during the pandemic, such as border duty patrolling, delivery of essential items, and overseeing the implementation of COVID-19 prevention protocols in the towns. This increases their likelihood of contracting the disease. This is consistent with the previous study, where the frontline workers were at a higher risk of contracting COVID-19, compared to the general community [38]. Additionally, our study also identified a significant risk of being symptomatic amongst the MDP workers. They are involved in shipping goods that come from the third country to Bhutanese carriers. Therefore, they are exposed to the risk of contracting the infection through the goods, and transporters ferrying the goods from India. Despite strict physical distancing protocols and PPE measures, MDP workers were amongst the first people to contract COVID-19 in Phuentsholing. Similar finding was also reported in another study where “drivers and transport workers” contributed to 18% of the COVID-19 burden in six Asian countries, including Hong Kong, Japan, Singapore, Taiwan, Thailand, and Vietnam [39]. It is, however, suggested that the coordination among states and governments is important to ensure that the control measures do not disrupt international trade and travel [40].

The results of this study should be interpreted in light of a few limitations. This is a retrospective study that is based on the limited data that was available at the isolation ward. Due to many missing variables, we were not able to include laboratory parameters in the analysis. The onset of symptoms could be subjected to the infection with different variants of SARS-CoV-2. Despite these limitations, our study identified findings that are similar to the other literature, and serves as a baseline to understand the epidemiological characteristic of COVID-19 in Bhutan.

## 9. Conclusions

In this study, we identified the host factors that increase the risk of developing COVID-19 symptoms in SARS-CoV-2 infection. MDP and frontline workers were at an increased risk of developing symptomatic COVID-19. Having at least one comorbid condition was associated with a longer length of stay in the isolation ward. Vaccination against COVID-19 was associated with a reduced risk of developing COVID-19 symptoms.

## Figures and Tables

**Figure 1 ijerph-18-10929-f001:**
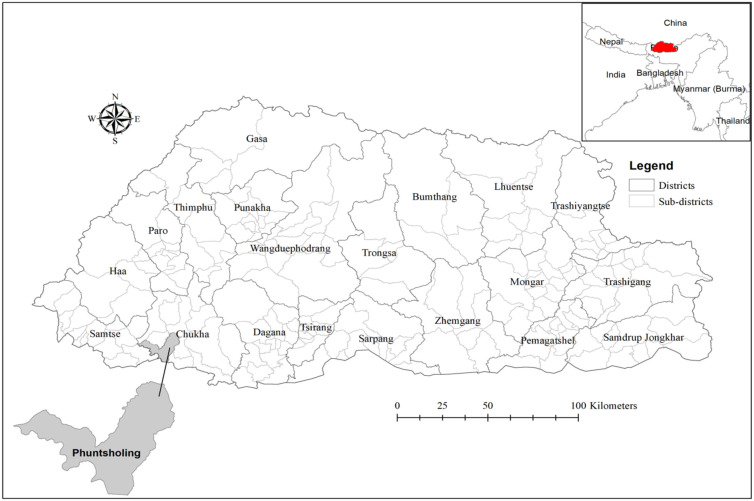
Political map of Bhutan showing Phuentsholing, the site of this study.

**Figure 2 ijerph-18-10929-f002:**
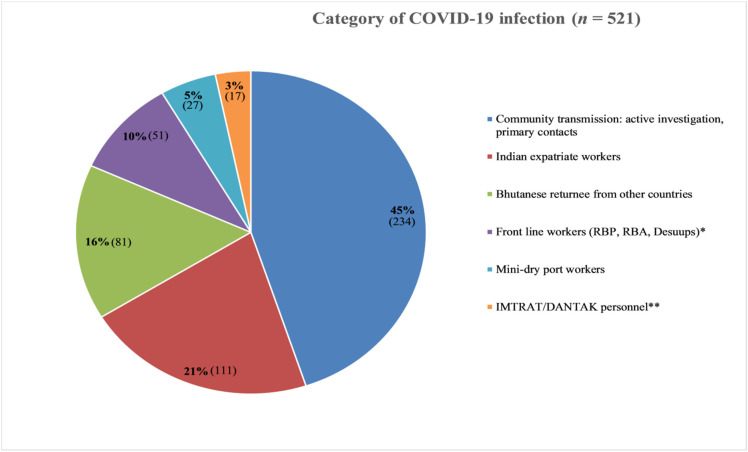
Categories of those infected with COVID-19 in Phuentsholing, Bhutan until 31 May 2021. * RBP- Royal Bhutan Police, RBA- Royal Bhutan Army, *Desuups*- volunteers. ** IMTRAT-Indian Military Training Team and DANTAK- Border Road Organization.

**Figure 3 ijerph-18-10929-f003:**
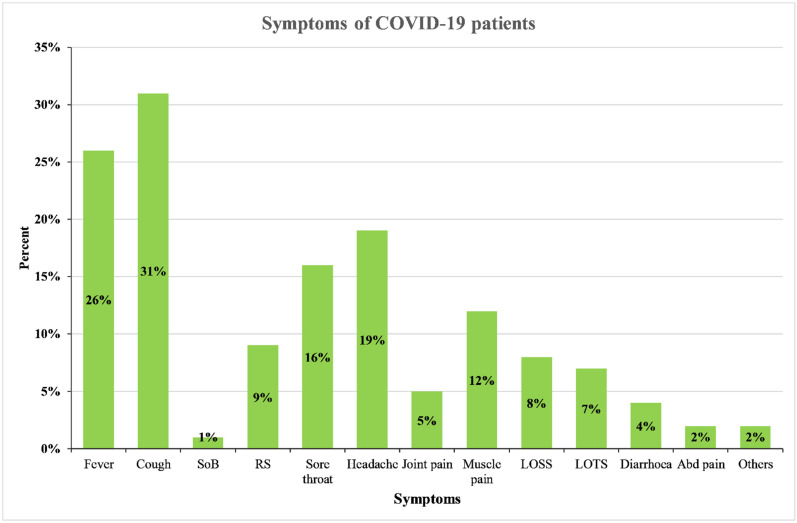
The proportion of different symptoms manifestations among COVID-19 patients. SOB- shortness of breath; RS- runny nose; LOSS- loss of smell sensation; LOTS- loss of taste sensation; Abd pain- abdominal pain; Others—chest pain, nausea, skin rashes.

**Figure 4 ijerph-18-10929-f004:**
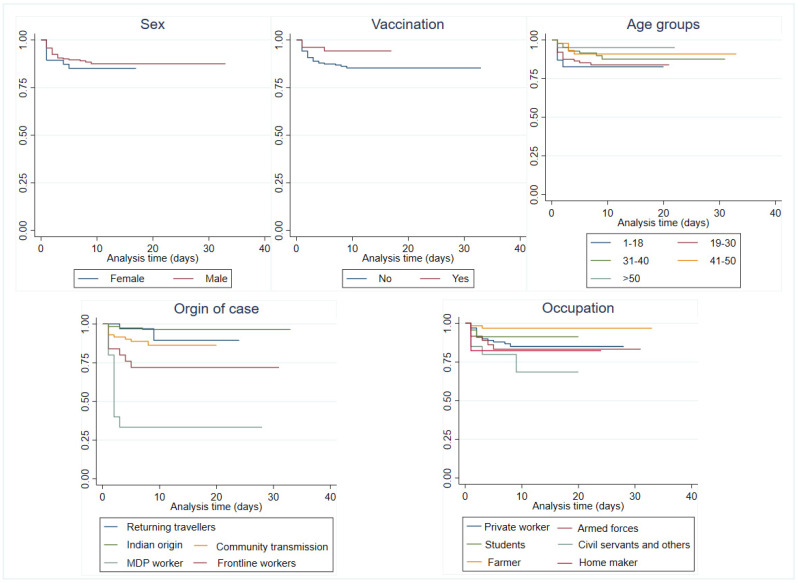
Kaplan-Meier survival estimates for the probability of developing symptomatic COVID-19 based on different covariates in the Phuentsholing municipality, Bhutan.

**Figure 5 ijerph-18-10929-f005:**
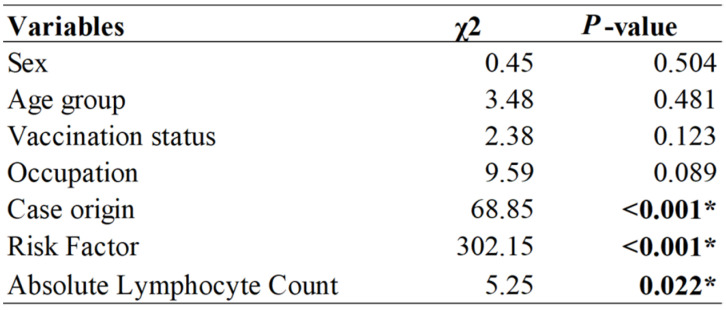
Kaplan–Meier estimator: comparison of the probability of occurrence of a disease (with symptoms) for patients with COVID-19 in the Phuentsholing municipality, Bhutan, for different variables. * significant at *p* < 0.05.

**Table 1 ijerph-18-10929-t001:** Characteristics of COVID-19 patients admitted to Phuentsholing isolation ward, Bhutan.

Characteristics	Number	Percent
Sex		
Male	368	70.6
Female	153	29.4
**Age group (years)**		
0–18	64	12.3
19–35	273	52.4
36–55	162	31.1
55–65	15	2.9
>65	7	1.3
**Occupation**		
Civil servant	16	3.1
Private/corporate/business	200	38.4
Student	78	15.0
Farmer	67	12.9
Armed force/uniformed personnel	65	12.5
Homemaker	54	10.4
Monk/nun	28	5.4
Other *	13	2.5
**Comorbidities and pregnancies**		
Uncomplicated diabetes mellitus	12	2.3
Pregnant women	4	0.7
Other **	3	0.5
**Symptomatic**		
No	224	43
Yes	290	56
**Onset of symptoms**		
Before admission	214	41.1
On admission day	47	9.0
During isolation	260	49.9
**Disease category among symptomatic (*n* = 290)**		
Mild disease	288	98.3
Severe pneumonia	5	1.7
Life-threatening anddeath	0	0
**Vaccination Status**		
Vaccinated	191	36.7
Not vaccinated	321	63.3

*—minor (age ≤6 years); **—obesity, history of pulmonary tuberculosis.

**Table 2 ijerph-18-10929-t002:** Haematological and serological data of COVID-19 patients in Phuentsholing.

Variable	Number	Mean	Std. Dev
Total white cell count	158	6.52 × 10^3^	2.149
Absolute lymphocyte count	158	1.88 × 10^3^	0.754
Platelets	131	239.72 × 10^3^	86.546
Number of days of isolation stay when antibodies were detected			
IgM *	30	17.37	6.054
IgG **	42	18.52	7.549

*—immunoglobulin M; **—immunoglobulin G against SARS-CoV-2.

**Table 3 ijerph-18-10929-t003:** Multivariable Cox proportional hazard model showing the adjusted hazard ratios of the different variables of COVID-19 patients in the Phuentsholing isolation ward.

Variables	Adjusted Hazard Ratio (95% Confidence Interval)	*p*-Value
**Sex**		
Male	Reference	
Female	0.95 (0.31, 2.84)	0.921
**Age group (years)**		
≤18	Reference	
19–30	3.56 (0.34, 37.12)	0.288
31–40	2.03 (0.19, 21.66)	0.559
41–50	1.41 (0.12, 16.03)	0.782
>50	0.28 (0.02, 3.62)	0.327
**Vaccination status**		
Non-vaccinated	Reference	
Vaccinated	0.23 (0.05, 0.98)	**0.047 ***
**Origin of cases**		
Returning travellers	Reference	
Frontline workers	14.90 (1.27, 174.09)	**0.031 ***
Indian origin	1.95 (0.14, 26.70)	0.616
Community transmission	8.33 (0.88, 78.79)	0.065
MDP worker	41.29 (4.24, 402.36)	**0.001 ***
**Occupation**		
Private worker	Reference	
Armed forces	1.26 (0.32, 4.92)	0.744
Student	1.36 (0.12, 15.58)	0.806
Farmer	0.73 (0.10, 5.36)	0.758
Civil servant and others	7.21 (0.87, 59.96)	0.067
Homemaker	2.33 (0.46, 11.77)	0.308

* significant at *p* < 0.05.

## Data Availability

Data supporting the results can be obtained upon request from Ministry of Health, Bhutan.

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
