# Peer review of "Survival Analysis of Symptomatic COVID-19 in Phuentsholing Municipality, Bhutan"

_ijerph, 2021, doi:10.3390/ijerph182010929_

Round 1
Reviewer 1 Report
Gyeltshen et al., documented the risk factors among COVID-19 patients Phuentsholing 10 COVID-19 Isolation ward, Bhutan. The manuscript is interesting and shared light on the disease as well contributing to the growing literature about the pandemic. The manuscript requires revision before accepting for publication. The grammar issues within the manuscript needs to be fixed when revising. Below are some comments to help improve the manuscript. Thank you
Line 9, 15 and 22. I will recommend that you remove introduction, method and conclusion from the abstract.
Line 9 please defined COVID -19
Line 16 please include the number and % of females
Line 19 Did the authors observed reinfection of vaccinated patients. If yes state?
Line 30. Please remove the full stop and link those two sentences
Line 33 please remove “to the pandemic”.
Figure 3 please consider removing the total number of patients according to the symptoms from the figure. The % are fine
Line 207 please report it like this 6.52 ± 2.149 x103 . Consider this throughout the manuscript and Table 2 as well. Consider deleting the min and max
Table 2 please states in the methods section how the “Haematological and serological analysis were conducted.
Line 219 the P values should come after “ no significant difference”.
Line 220 significant difference??? Whats the p value
Line 221 “origin of COVID-221 19” please rephrase
Line 228 what do you mean by case origin?
Author Response
Dear Reviewer 1,
Thank you for your comments. Please see the attachment for responses to your comments.
Thank you.

Reviewer 2 Report
The authors in this retrospective study have studied a cohort of COVID-19 positive patients travelling Phuntsholing point of entry which included both Bhutanese and Indian nationals. The aim of this study was to identify risk factors contributing to development of symptoms in COVID-19 positive patients. A sample size of 521 patients was used for this study constituting 368 (70.6%) of males with a mean age of 32 years with atleast one symptom. The major findings of this study were that vaccinated individuals were less likely to develop symptoms compared to those who were not vaccinated. MDP workers and front-line workers were more likely to develop symptoms. The study performed used rigorous and appropriate statistical analysis to draw conclusions from this study.
Author Response
Thank you very much and we are pleased to hear your comments. The minor English spell check was carried out as you suggested.
Reviewer 3 Report
In their manuscript entitled, “Survival Analysis of Symptomatic COVID-19 in Phuntsholing Municipality, Bhutan”, Gyeltshen et al. analyzed retrospectively 521 patients in Bhutan from May 2020 to May 2021 who were placed in the COVID isolation ward and tested positive for SARS-CoV-2. Risk factors were assessed for developing COVID, and the authors found that what put an individual at most risk for disease was being a front line or port worker, and most importantly being unvaccinated. In addition, the young and economically active groups, which likely are the most mobile and interactive, were affected the most.
Overall, the paper is clearly written and gives a nice summary of the COVID outbreak and risk factors in Bhutan. With COVID spread world-wide, it is helpful to gather as much information to share and emphasize that vaccinations do work and increase your chance of survival. Only a few comments below:
Line 55: I would give more details of what the Covishield vaccine is… not mRNA based, is a replication defective chimpanzee adenoviral vector expressing the SARS-CoV-2 spike glycoprotein, also known as ChAdOx1, etc.
Line 55: Also give more information on Covishield’s efficacy in clinical trials.
Line 58: Provide more information/references on Covishield’s effectiveness against the delta (and/or other) variants, especially since it is stated that >80% of cases in Bhutan were delta-related.
Define vaccinated in your study. Does that mean at least one dose, or 2 doses of Covisheild? I believe at the end of the discussion you finally mention that actually the patients in the study only had one dose; maybe state that early on in the paper for the reader as well. This could influence survival rates. Vaccinated patients were 77% less likely to develop COVID systems, and this maybe would go up if patients had 2 doses.
Were there any rare cases of Guillain-Barre syndrome in the vaccinated patients?
For comorbidities associated with COVID, do you have information on whether the patients smoked? Heart disease/hypertension?
Line 273: I feel using the wording “inconsistent with previous studies” is a little harsh. All studies from around the world have some variability. I would reword the sentence to say “2-3 fold lower than other studies”.
Figure 4: The wording in the “legend boxes” overlaps and is hard to read. You might be able to take for example the word “occupation” and place it in the graph itself so that the box below will now fit each descriptive occupation better. Can do the same for all graphs.
Author Response
Dear Reviewer 3,
Thank you for your comments. Please see the attachment for the responses to your comments.
Thank you.
